# Prompt-based Depth Pruning of Large Language Models

**Juyun Wee** [* 1]  **Minjae Park** [* 1]  **Jaeho Lee** [1]

## Abstract

Depth pruning aims to reduce the inference cost of a large language model without any hardware-specific complications, by simply removing several less important transformer blocks. However, our empirical findings suggest that the importance of a transformer block may be highly task-dependent—a block that is crucial for a task can be removed without degrading the accuracy on another task. Based on this observation, we develop a dynamic depth pruning algorithm, coined PuDDing (Prompt-routed Dynamic Depth Pruning), which determines which blocks to omit from the model based on the input prompt. PuDDing operates by training a lightweight router to predict the best omission set among a set of options, where this option set has also been constructed in a data-driven manner. Empirical results on commonsense reasoning benchmarks demonstrate that PuDDing effectively accelerates the inference language models, and achieves better on-task performance than static depth pruning baselines.

**Project Page**: jwee01.github.io/PuDDing
**Code**: github.com/tada0347/PuDDing

## 1. Introduction

Recent advances in large language models (LLMs) have achieved remarkable success in a wide range of natural language processing tasks (Brown et al., 2020; Touvron et al., 2023; Dubey et al., 2024). However, significant computational requirements of LLMs pose challenges in resource-constrained environments, limiting their practicality. For example, LLaMA-3.3-70B needs 140GB of RAM to be loaded in bf16, which is often too big for memory-constrained local devices. Thus, reducing the model size is essential to make LLMs feasible for on-device applications.

Depth pruning is a versatile model compression technique

[*]Equal contribution  [1]POSTECH. Correspondence to: Jaeho Lee <jaeho.lee@postech.ac.kr>.

*Proceedings of the $42^{nd}$ International Conference on Machine Learning*, Vancouver, Canada. PMLR 267, 2025. Copyright 2025 by the author(s).

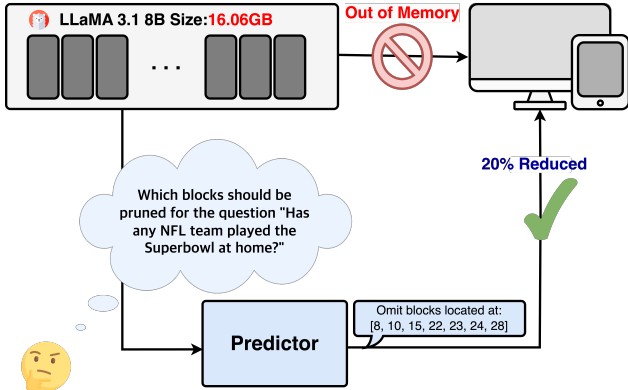

*Figure 1.* The general framework of prompt-based depth pruning. Given some query from the user, the goal is to identify which layers from an LLM can be omitted, so that one can make accurate prediction on low-memory consumer devices.

that is particularly effective for on-device scenarios (Song et al., 2024; Kim et al., 2024). Such methods simply remove several transformer blocks (which we call "omission set") from the pretrained model, based on some measures of block importance computed using a small amount of calibration samples. As everything is identical except for the number of blocks, the pruned model is suitable to be deployed on any hardware without tailored supports on low-precision (*e.g.*, integer cores) or fine-grained sparsity (*e.g.*, 2:4 sparsity). Furthermore, as there is no extensive training involved, depth pruning can be easily done in a device-by-device manner for deployment on various devices.

A key limitation of typical depth pruning algorithms is that their pruning decision is ***static***, *i.e.*, the same omission set is removed regardless of the query given to the model. While this choice allows one to save storage (*e.g.*, flash drives) by discarding the pruned parameters at the local device, it sacrifices the ability to adapt to various downstream tasks. Indeed, our empirical observations show that pruning some transformer blocks in an LLM may incur significant accuracy degradation on certain tasks, while being highly unnecessary for other tasks (see Section 3).

Can we make dynamic depth pruning decisions to improve the performance on various tasks? This question has not been well studied yet, especially in the context of on-device

inference. A recent line of work develops effective ***dynamic token routing*** mechanisms to save training/inference computation by processing each token with a limited number of transformer blocks (Raposo et al., 2024; Wang et al., 2024). However, such methods require all parameters to be loaded on high-speed memories (*e.g.*, on-GPU memory); thus, the methods are appropriate for large-scale server clusters, not for on-device inference with memory constraints.

**Contribution.** To overcome the limitations, we develop a new ***prompt-based depth pruning*** approach (Section 4): In the pre-fill stage, based on the prompt given from the user, a limited number of transformer blocks are selected and loaded to the on-device RAM from the storage drive. This approach does not require a large memory to hold all parameters or highly repeated per-token routing, and thus can effectively accelerate inference on low-memory devices.

A naïve way to achieve this goal might be to conduct conventional static depth pruning at each inference, using the given prompt as calibration samples. However, this approach incurs a large latency in running static pruning algorithms in every inference. Furthermore, such a method is likely to fail making a good pruning decision due to the shortage of calibration data, especially in single-batch inference cases common in on-device scenarios.

To this end, we propose a *training-based* method for the prompt-based depth pruning of large langauge models (Section 5). Our method, coined Prompt-routed Dynamic Depth Pruning (PuDDing), works in two steps.

1. *Candidate omission set generation.* We construct a small yet diverse and performant family of omission sets. This is done by drawing multiple splits of calibration data from various task dataset, and then finding an omission set which achieves low loss on each split; here, we use a newly developed task-centric loss instead of perplexity.

2. *Router training.* We train a lightweight router which predicts the appropriate omission set from the given prompt. This is done by generating a training dataset consisting of prompt-loss pairs for each omission set, and training the model to predict the loss from the prompt; routing can be done by choosing the minimum-loss option.

Empirically, we find that the proposed PuDDing enjoys a clear advantage over static depth pruning algorithms, achieving more than 4%p accuracy increase on zero-shot commonsense reasoning tasks (Section 6). At the same time, as the algorithm uses the router only once per each prompt, PuDDing enjoys over 1.2× generation speedup over the dense model, similar to the static depth pruning algorithms.

Our key contributions can be summarized as follows:

- Our observations reveal that optimal depth pruning decisions may be highly depend on the task given at hand,

underscoring the need for task-dependent depth pruning.
- We consider the task of prompt-based depth pruning for the first time (to our knowledge), and propose a training-based strategy as a solution.
- Comparing with static depth pruning algorithms, our algorithm achieves a much higher zero-shot accuracies on various tasks, while being competitive in terms of the computational efficiency.

*Table 1.* A high-level comparison of the proposed prompt-based depth pruning framework with related depth pruning approaches: Static depth pruning and dynamic token routing.

| | Task Adaptive | Peak Memory | Routing |
|---|---|---|---|
| Static Pruning (Song et al., 2024; Kim et al., 2024) | ✗ | Sparse | - |
| Token Routing (Raposo et al., 2024; Wang et al., 2024) | ✓ | Dense | Per token |
| Prompt-based depth pruning (this paper) | ✓ | Sparse | Per prompt |

## 2. Related Work

In this section, we provide an in-depth comparison of the proposed framework against existing depth and width sparsity frameworks. See Table 1 for a concise summary.

### 2.1. Static Depth Pruning

Static depth pruning methods select and remove unnecessary blocks from a pretrained LLM using various proxy metrics to measure the importance of the blocks. ShortGPT (Men et al., 2024) measures the block importance using the expected cosine similarity between the input and output activations of the block; a block that does not change the direction of the activation is deemed unnecessary. Shortened-LLaMA (Kim et al., 2024) directly measures the perplexity drop after removing each transformer block, and SLEB (Song et al., 2024) combines this idea with an iterative pruning.

Several recent works also focus on layer-level depth pruning, instead of removing an entire transformer block. In particular, Siddiqui et al. (2024), He et al. (2024) discover that pruning out self-attention layers have a much less significant impact than removing the feed-forward layers.

Unlike these works, this paper aims to perform dynamic depth pruning using the prompts for the downstream tasks; to account for this difference, we design and use new likelihood-based metrics to measure the block importance.

### 2.2. Dynamic Token Routing

Inspired by the success of mixture-of-experts (Jacobs et al., 1991; Fedus et al., 2022), several recent works have developed mechanisms to route tokens through only a fraction of all transformer blocks. Mixture-of-Depth (Raposo et al., 2024) adopts the depth sparsity during the training phase with a jointly trained router, to reduce the training cost of

LLMs. Here, the trained router can also be used at inference. D-LLM (Wang et al., 2024) trains a router that can be applied on pretrained LLMs to reduce their inference cost.

Our approach differs from both of these works in the sense that it needs only a limited number of transformer blocks active for a single input query (or prompt); the routing is conducted once per input prompt, not per token.

### 2.3. Contextual Sparsity

Our work is most closely related to the idea of *contextual sparsity*, where a lightweight router selects an input-dependent subnetwork at inference time without updating the base weights. In the context of width pruning, prior works—Deja Vu (Liu et al., 2023), ShadowLLM (Akhauri et al., 2024), Sirius (Zhou et al., 2024), and CATS (Lee et al., 2024)—have demonstrated that context-aware routing can be done with minimal or no degradation in task performance. **PuDDing extends this paradigm to *depth* pruning for the first time**: instead of skipping neurons or channels, our router decides which entire transformer blocks to omit. This preserves the original matrix shapes and avoids hardware mismatches often caused by width pruning.

## 3. A Motivating Observation

Before describing the proposed framework, we briefly describe a motivating observation which demonstrate that:

> The importance of a transformer block in a language model may be highly ***task-dependent***.

**Setup.** To show this point, we have compared the zero-shot accuracies of the LLMs whose omission sets differ by a single transformer block. More concretely, we compare the performance of an omission set $(b_1, b_2, \ldots, b_{k-1}, b_k)$ to another omission set $(b_1, b_2, \ldots, b_{k-1}, \tilde{b}_k)$, on the LLaMA 3.1-8B model. Here, we have used the SLEB (Song et al., 2024) to generate an omission set, and then replaced a single block to get another one. Then, we observe the impact of such replacement on three commonsense reasoning tasks: BoolQ, PIQA, and WinoGrande.

**Result.** Figure 2 illustrates our findings. We observe that pruning out block 29 instead of block 30 has a two-sided impact: On BoolQ, the change makes a dramatic drop in accuracy ($62.2\% \rightarrow 38.0\%$, $62.5\% \rightarrow 37.9\%$). However, on PIQA and WinoGrande, we observe a slight accuracy boost. This phenomenon suggests that the block 29 may contain more knowledge relevant to answering BoolQ questions, while 30 may be more knowledgeable about PIQA and WinoGrande. This observation highlights the need to consider task variability during the selection of the omission set. To formally address such need, this paper considers an

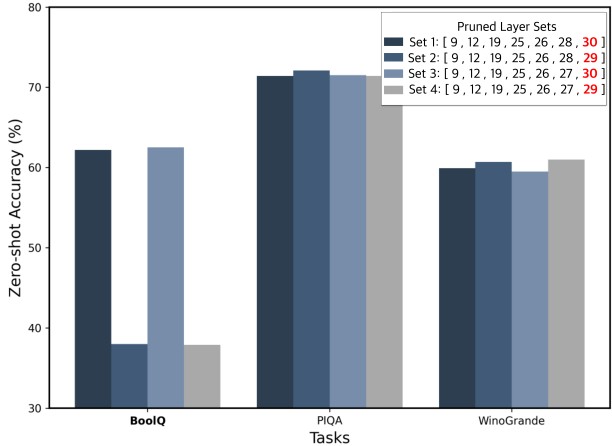

*Figure 2.* The impact of pruning the transformer block 29 vs. block 30. On the BoolQ dataset, pruning the block 29 instead of block 30 incurs a dramatic performance degradation, with over 20%p drop. On the other hand, on PIQA and WinoGrande, the accuracy does not change much, or even increases.

inference of task information from the prompt.

## 4. Problem Description

Inspired by the observations in Section 3, we now formalize the problem of prompt-based depth pruning.

In a nutshell, given some pretrained LLM and a prompt, the goal of the prompt-based depth pruning is to designate which transformer blocks should be removed from the model to generate the most accurate response to the prompt.

More concretely, let $\mathbf{x}$ be the prompt given to the model, and let $\mathbf{W} = (W_1, \ldots, W_d)$ be the weight parameters of a pretrained LLM consisting of $d$ transformer blocks, with $W_i$ indicating the weights of the $i$th block. The prediction quality of this language model is measured by the expected loss between the model output and the ground-truth, *i.e.*,

$$L(\mathbf{W}) := \mathbb{E}[\ell((\mathbf{x}, \mathbf{y}); \mathbf{W})], \tag{1}$$

where $\ell((\cdot, \cdot); \mathbf{W})$ is some loss function which also encapsulates the generative procedure of language model with parameter $\mathbf{W}$ (*e.g.*, perplexity). In static depth pruning, the goal is to find which blocks to prune from the given LLM. More formally, define **omission set** as a(n unordered) set of transformer block indices

$$\mathbf{b} = \{b_1, b_2, \ldots, b_k\} \subseteq \{1, 2, \ldots, d\}, \tag{2}$$

which designates which blocks will be omitted from the target LLM. Then, let $\mathbf{W}^{\backslash \mathbf{b}}$ be a sequence of $d - k$ weights, with $b_i$th block eliminated from the $\mathbf{W}$. Then, the static

depth pruning aims to solve the minimization

$$\min_{\mathbf{b}:|\mathbf{b}|\geq k} L(\mathbf{W}^{\backslash \mathbf{b}}), \tag{3}$$

given the depth constraint $k$ designated by the operational constraints, such as the desired latency or the peak memory.

**Prompt-based Depth Pruning.** The problem of prompt-based depth pruning can be described as optimizing the omission set as a function $\hat{\mathbf{b}}(\mathbf{x})$, *i.e.*, solving

$$\min_{\hat{\mathbf{b}}(\cdot)} \mathbb{E}\big[\ell((\mathbf{x},\mathbf{y});\mathbf{W}^{\backslash \hat{\mathbf{b}}(\mathbf{x})})\big], \tag{4}$$

$$\text{subject to} \quad \mathbf{Pr}\big[|\hat{\mathbf{b}}(\mathbf{x})| \geq k\big] = 1.$$

Note that we are constraining the omission set to have the cardinality greater than $k$ for all $\mathbf{x}$. In other words, the pruned model should always have $d - k$ or less blocks. This is because we mainly consider the peak memory constraint, *i.e.*, the RAM cannot hold more than $d - k$ blocks. Otherwise, one can consider a slightly modified version of the problem (4) with a probabilistic constraint.

## 5. Method

We now formally describe the proposed PuDDing (Prompt-routed Dynamic Depth Pruning)—an algorithm to train a router $\hat{\mathbf{b}}(\cdot)$ for the prompt-based depth pruning.

In a nutshell, PuDDing operates in two steps:

1. Generating candidate omission sets using the prompt-answer dataset collected from various tasks (Section 5.1)
2. Training a router to predict the best option among the candidate omission sets (Section 5.2)

During the inference phase, the given prompt is fed to the router, which predicts which omission set (among the candidates) one should use for the given prompt. Then, the model parameters are loaded from the storage to the high-speed memory to constitute the depth-pruned LLM (see Figure 3).

We note that this classification-based approach is in contrast with the approach of dynamic token routing (Wang et al., 2024), where one makes yes/no decisions for omitting each block in a sequential manner; this change is to make the router training easier and generalizable.

### 5.1. Candidate Omission Set Generation

The first step is to generate a candidate pool of omission sets. That is, we generate a family of omission sets

$$\mathcal{B} = \{\mathbf{b}_1, \ldots, \mathbf{b}_m\}, \tag{5}$$

which will be used as the codomain of the router $\hat{\mathbf{b}}(\cdot)$, which will simply be an $m$-class classifier.

Desirable properties of the candidate set $\mathcal{B}$ are as follows:

- Coverage: For any realistic prompt-answer pair $(\mathbf{x},\mathbf{y})$ from a wide range of tasks, the set $\mathcal{B}$ should contain at least one $\mathbf{b}_i$ with a small loss $\ell(\mathbf{y}, f(\mathbf{x};\mathbf{W}^{\backslash \mathbf{b}_i}))$.
- Cardinality: The number of omission sets $m$ should be sufficiently small, so that one can train a nice predictor for $\mathcal{B}$ with a limited number of samples.

To obtain these properties, we adopt the following strategy: First, we collect $t$ calibration datasets $D_1, \ldots, D_t$ on a diverse set of downstream tasks. Then, on each calibration dataset, we select the omission set that minimizes some loss criterion, *i.e.*, solve

$$\mathbf{b}_i = \arg\min_{\mathbf{b}} \mathbb{E}_{D_i}\big[\ell(\mathbf{y}; f(\mathbf{x};\mathbf{W}^{\backslash \mathbf{b}}))\big]. \tag{6}$$

Here, the minimization is done in a greedy manner, similar to Song et al. (2024). We apply $l$ different loss criteria on each calibration dataset to get $m = t \times l$ omission sets.

**Losses.** As the loss function, we use new task-focused variants of the perplexity loss, which we call the ***task likelihood losses***. The perplexity measures the fluency of the generated sentences by measuring the average log-likelihood losses over the whole sequence. That is, for a sample sentence $\mathbf{z} = (z_1, z_2, \ldots, z_T)$, the perplexity is

$$\text{ppl}(\mathbf{z};\mathbf{W}) = \exp\left(-\frac{1}{T}\sum_{i=1}^{T}\log p_i(z_i|z_{<i};\mathbf{W})\right), \tag{7}$$

where $p_i(\cdot|\cdot;\mathbf{W})$ denotes the conditional generative probability of the target language model with parameters $\mathbf{W}$, at the $i$th token. We modify this loss to measure the likelihood only the sequence that matters for on-task performance. That is, if the given datum $\mathbf{z}$ can be broken down into the prompt and answer pair:

$$\mathbf{z} = (\mathbf{x},\mathbf{y}) = (\underbrace{z_1, \ldots, z_S}_{=\mathbf{x}}, \underbrace{z_{S+1}, \ldots, z_T}_{=\mathbf{y}}), \tag{8}$$

then we can define the *task likelihood (tl)* loss as:

$$\text{tl}(\mathbf{z};\mathbf{W}) = -\frac{1}{T-S}\sum_{i=S+1}^{T}\log p_i(z_i|z_{<i};\mathbf{W}). \tag{9}$$

In addition, we also consider the *task likelihood difference* (tld) loss, which is defined as follows: In many tasks, the answer choices are limited (*e.g.* "true" or "false"). In such cases, we can also use the likelihood difference of the correct and wrong answers, i.e.,

$$\text{tld}(\mathbf{z};\mathbf{W}) = \text{tl}((\mathbf{x},\mathbf{y});\mathbf{W}) - \text{tl}((\mathbf{x},\mathbf{y}^{\text{wrong}});\mathbf{W}), \tag{10}$$

where $\mathbf{y}^{\text{wrong}}$ denotes the wrong version of the answer. We use both $\text{tl}(\cdot)$ and $\text{tld}(\cdot)$ as our loss criteria.

We note that the task likelihood losses Equations (9) and (10) is different from the perplexity (Equation (7)), in the sense that we do not exponentiate the values. We use this version as it empirically works better than the exponentiated one.

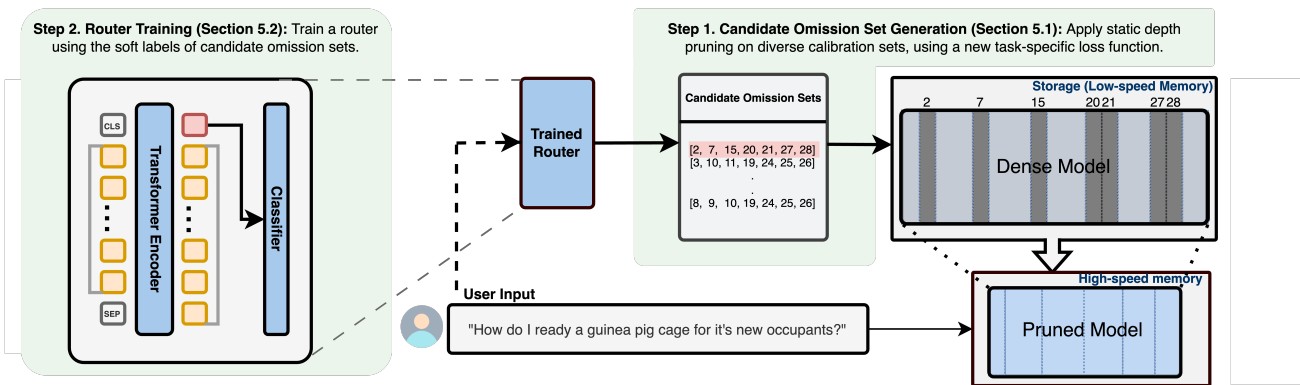

*Figure 3.* A visual overview of the proposed pipeline. Whenever the prompt is given from the user, a trained router predicts which set of blocks can be omitted with minimal loss, among the small number of candidate omission sets. Then, the LLM transformer blocks are loaded from the storage (*e.g.*, flash drive) to the high-speed memory (*e.g.*, GPU RAM) except for the omitted blocks, saving the time and energy for the data communication. Finally, the depth-pruned model operates and generates the text.

## 5.2. Router Training

After generating the candidate omission set $\mathcal{B}$, we train a router that maps the given prompt to the best omission set. Roughly, this is done by first constructing a soft-labeled dataset with task-specific datasets and then training a BERT-based router on the constructed dataset (Devlin et al., 2019)

**Dataset Construction.** To construct the training dataset, we first collect various prompt-answer pairs from the task datasets, similarly to the calibration datasets in Section 5.1. Then, for each sample, we compute the task likelihood losses on all omission sets, and store them as a label vector. That is, each datum inside the dataset takes the form $(\mathbf{x}_i, \mathbf{s}_i)$, where $\mathbf{x}_i$ is the prompt and the $\mathbf{s}_i$ is a length-$m$ vector with

$$\mathbf{s}_i = \big(\mathrm{tl}((\mathbf{x}_i, \mathbf{y}_i); \mathbf{W}^{\backslash \mathbf{b}_1}), \ldots, \mathrm{tl}((\mathbf{x}_i, \mathbf{y}_i); \mathbf{W}^{\backslash \mathbf{b}_m})\big). \quad (11)$$

Note that we no longer need to store the correct answers $\mathbf{y}_i$.

**Router training.** We train a router to accurately predict the label vector $\mathbf{s}$ given the input prompt $\mathbf{x}$, for all samples in this dataset. That is, we train a function $\hat{\mathbf{s}} = f(\mathbf{x})$ such that $\hat{\mathbf{s}} \approx \mathbf{s}$ holds. We use the MSE loss

$$\mathrm{MSE}(\mathbf{s}, \hat{\mathbf{s}}) = \|\mathbf{s} - \hat{\mathbf{s}}\|_2^2 \quad (12)$$

to train the router. At inference, we will select the omission set with the minimum-achieving index of the predicted $\hat{\mathbf{s}}$.

**Router architecture.** We use a lightweight transformer-based encoder as our router. More specifically, we insert a single linear layer on pretrained BERT-base (Devlin et al., 2019), and jointly fine-tune during the training. While this router has more parameters ($\sim$110M) than typical routers that are used for dynamic token routing—such as D-LLM (Wang et al., 2024) which uses 2-layer MLP—the computational cost is bearable as we route only once per prompt. In our experiments, the routing cost typically takes up around $2 - 4\%$ of the total pre-fill cost.

## 6. Experiment

We now empirically validate the performance of the proposed algorithm, PuDDing, on zero-shot tasks against the static depth and width pruning baselines.

### 6.1. Experimental Setup

**Models.** We evaluate the proposed method on compressing three popular open-weight language models. As the main model, we use the LLaMA-3.1 model with 8B parameters (Dubey et al., 2024). In addition, we evaluate on two language models: Vicuna 1.5 with 7B (Chiang et al., 2023), and OPT with 6.7B parameters (Zhang et al., 2022). We use these models for two reasons. First, the models have an appropriate scale for on-device deployments. Second, all three models consist of 32 transformer blocks, and thus can be compared with the same sparsity criterion.

**Baselines.** We mainly compare against four recent static depth and width pruning baselines with open source.

- SLEB (Song et al., 2024): A depth pruning baseline that iteratively selects the omission set based on perplexity.
- Shortend LLaMA (Kim et al., 2024): A depth pruning algorithm which selects the omission set one-shot; here, we compare with the version that uses perplexity as the loss criterion and does not apply LoRA.
- FLAP (An et al., 2024): A retraining-free width pruning algorithm based on structural fluctuation metric.
- SliceGPT (Ashkboos et al., 2024): Another width pruning algorithm based on principal component analysis.

In addition, we also compare with "SLEB per prompt," which is simply SLEB which is conducted by using each given prompt as the calibration data. As this option does not work well in general, and requires a long inference time, we

*Table 2.* Zero-shot accuracy comparisons of PuDDing against baseline compression algorithms on LLaMA-3.1 8B on commonsense reasoning tasks. The best performances are marked in **bold**, and the runner-up is marked with underline.

| Method | Structure | Pruned Blocks (Sparsity) | Per-task Accuracies | | | | | | Average Acc. (%) |
| --- | --- | --- | --- | --- | --- | --- | --- | --- | --- |
| | | | Arc-C | Arc-E | BoolQ | HellaSwag | PIQA | WinoGrande | |
| Dense | - | 0 | 53.50 | 81.52 | 82.20 | 78.81 | 79.98 | 73.40 | 74.90 |
| FLAP | Width | - (20%) | 26.54 | 46.80 | 62.32 | 46.93 | 64.58 | 58.56 | 50.96 |
| SliceGPT | Width | - (20%) | 34.30 | 65.15 | 44.52 | 60.55 | 73.67 | 56.43 | 50.28 |
| SLEB | Depth | 7 (>21%) | 34.90 | 66.25 | 49.11 | 61.60 | 74.37 | 57.22 | 57.24 |
| SLEB per prompt | Depth | 7 (>21%) | 33.44 | 50.59 | 57.95 | 53.57 | 63.44 | 56.51 | 52.58 |
| Shortened LLaMA | Depth | 7 (>21%) | 34.30 | 65.15 | 44.52 | 60.55 | 73.67 | 56.43 | 55.77 |
| PuDDing (Ours) | Depth | 7 (>21%) | 41.47 | 67.09 | 62.02 | 62.92 | 73.94 | 64.16 | **61.93** (+4.69) |
| FLAP | Width | - (15%) | 33.19 | 60.02 | 69.45 | 58.18 | 71.16 | 61.88 | 58.98 |
| SliceGPT | Width | - (15%) | 32.59 | 59.60 | 49.82 | 58.59 | 67.14 | 64.56 | 55.38 |
| SLEB | Depth | 5 (>15%) | 39.59 | 70.58 | 58.17 | 67.16 | 75.63 | 63.77 | 62.48 |
| Shortened LLaMA | Depth | 5 (>15%) | 40.78 | 69.11 | 60.67 | 67.46 | 76.28 | 64.09 | 63.07 |
| PuDDing (Ours) | Depth | 5 (>15%) | 42.32 | 72.39 | 65.11 | 67.28 | 75.79 | 65.35 | **64.71** (+1.64) |
| FLAP | Width | - (10%) | 36.43 | 66.20 | 69.69 | 63.29 | 74.10 | 66.61 | 62.72 |
| SliceGPT | Width | - (10%) | 38.14 | 68.90 | 63.67 | 65.47 | 70.78 | 66.30 | 62.21 |
| SLEB | Depth | 3 (>9%) | 45.73 | 76.01 | 68.93 | 71.96 | 77.53 | 68.98 | 68.19 |
| Shortened LLaMA | Depth | 3 (>9%) | 38.57 | 69.91 | 69.72 | 71.28 | 77.31 | 67.48 | 65.71 |
| PuDDing (Ours) | Depth | 3 (>9%) | 48.98 | 77.02 | 70.18 | 73.26 | 77.20 | 68.11 | **69.13** (+0.94) |

*Table 3.* Zero-shot task accuracy comparison on LLaMA 3.1 8B, OPT 6.7B, and Vicuna 1.5 7B. The best performances are marked in **bold**, and the runner-up is marked with underline. We have applied 20% sparsity (*i.e.*, pruned seven blocks).

| Method | LLaMA 3.1 8B | OPT 6.7B | Vicuna 1.5 7B |
| --- | --- | --- | --- |
| Dense | 74.90 | 62.51 | 70.49 |
| FLAP | 50.96 | 46.68 | 51.45 |
| SliceGPT | 50.28 | 55.45 | 59.11 |
| SLEB | 57.24 | 56.55 | 58.68 |
| Shortened LLaMA | 55.77 | 54.58 | 59.78 |
| PuDDing (Ours) | **61.93** | **58.37** | **60.01** |

only compare on a limited number of scenarios.

**Dataset: Evaluation.** We evaluate on the test splits of six zero-shot commonsense reasoning tasks: ARC-Challenge and ARC-Easy (Clark et al., 2018), HellaSwag (Zellers et al., 2019), PIQA (Bisk et al., 2020), WinoGrande (Sakaguchi et al., 2020), and BoolQ (Clark et al., 2019).

**Dataset: Calibration for Baselines.** For the baseline algorithms, we have used the calibration data designated in the original papers. For SLEB, FLAP, and SliceGPT, we have used the WikiText-2 (Merity et al., 2022). For Shortened LLaMA, we have used the BookCorpus (Zhu et al., 2015).

**Training.** To generate the candidate omission set for our algorithm, we have used 128 randomly drawn samples from the training splits of five zero-shot commonsense reasoning

tasks: ARC-Challenge, ARC-Easy, HellaSwag, PIQA, and WinoGrande. That is, we use total 10 omission sets (as we use two different losses). For training the router, we have used the full training splits. BoolQ dataset has been left out in order to evaluate the generalization to unseen sets. The router has been trained with AdamW with learning rate $10^{-5}$, weight decay 0.01, and batch size 32 for 10 epochs, with 500 warm-up steps. Also, for WinoGrande dataset, we use a tailored data pre-processing procedure; we describe this in detail in Appendix A.

**Hardware.** We have mainly used NVIDIA RTX 6000 Ada for evaluation and training. In addition, we have used cloud instances of NVIDIA A100 for evaluation.

### 6.2. Main Experiment

Table 2 provides a comparison of zero-shot accuracies of the model compression methods, on LLaMA-3.1 8B model. From the table, we observe that PuDDing achieves the highest average accuracy on all sparsity levels tested. Especially when 7 blocks have been pruned (over 20% sparsity), the improvement over the best baselines is almost 3%p.

An interesting observation is the poor performance of "SLEB per prompt," which measures which block to remove on the fly, by using the given prompt as a calibration dataset. In fact, the performance is worse than the vanilla SLEB. We hypothesize that this is because a single prompt usually does not contain enough information to work as a good calibration data. Our training-based strategy circumvents such

*Table 4.* Zero-shot accuracy comparisons of PuDDing vs. other depth pruning methods on LLaMA-3.1 8B, with LoRA finetuning. The best performances are marked in **bold**, and the runner-up is marked with underline.

| Method | Pruned Blocks (Sparsity) | Per-task Accuracies | | | | | | Average Acc. (%) |
|---|---|---|---|---|---|---|---|---|
| | | Arc-C | Arc-E | BoolQ | HellaSwag | PIQA | WinoGrande | |
| Dense | 0 | 53.50 | 81.52 | 82.20 | 78.81 | 79.98 | 73.40 | 74.90 |
| SLEB + LoRA | 7 (>21%) | 45.39 | 74.92 | 69.05 | 70.92 | 78.35 | 64.40 | 67.17 |
| Shortened LLaMA + LoRA | 7 (>21%) | 43.52 | 74.07 | 63.88 | 71.74 | 78.35 | 63.85 | 65.90 |
| LLM-Streamline (w/ fine-tune) | 7 (>21%) | 44.80 | 70.12 | 70.06 | 67.15 | 72.63 | 71.74 | 66.08 |
| PuDDing + LoRA (Ours) | 7 (>21%) | 45.39 | 75.34 | 71.96 | 71.58 | 77.26 | 66.54 | **68.01** (+0.84) |

*Table 5.* Accuracy comparisons of PuDDing vs. other depth pruning methods on LLaMA-3.1 8B in the unseen tasks that require more complicated reasoning. The best performances are marked in **bold**, and the runner-up is marked with underline.

| Method | Pruned Blocks | OpenBookQA | MathQA | MMLU | PubMedQA | SciQ |
|---|---|---|---|---|---|---|
| Dense | 0 | 44.60 | 39.53 | 63.49 | 75.80 | 96.00 |
| SLEB | 7 | 36.00 | 25.19 | 23.76 | 56.40 | 89.20 |
| Shortened LLaMA | 7 | 34.20 | 25.76 | 26.78 | 52.60 | 89.20 |
| PuDDing | 7 | **36.40** | **27.20** | **39.00** | **60.00** | **92.70** |

*Table 6.* Wall clock inference speed of the PuDDing-compressed LLaMA-3.1 8B evaluated on NVIDIA A100 and RTX 6000 Ada.

| A100 | Pre-fill (TTFT) | | | Pre-fill + Generation | | |
|---|---|---|---|---|---|---|
| Prompt Length | 128 | 256 | 512 | 128 | 128 | 128 |
| Gen. Length | 1 | 1 | 1 | 128 | 256 | 512 |
| Dense | 0.137s | 0.251s | 0.505s | 3.296s | 6.634s | 13.595s |
| PuDDing | 0.109s | 0.201s | 0.393s | 2.694s | 5.375s | 11.024s |
| →Router | +0.004s | + 0.005s | +0.008s | +0.004s | +0.004s | +0.004s |
| Speedup | 1.21× | 1.22× | 1.23× | 1.22× | 1.23× | 1.23× |

| RTX 6000 Ada | Pre-fill (TTFT) | | | Pre-fill + Generation | | |
|---|---|---|---|---|---|---|
| Prompt Length | 128 | 256 | 512 | 128 | 128 | 128 |
| Gen. Length | 1 | 1 | 1 | 128 | 256 | 512 |
| Dense | 0.008s | 0.171s | 0.323s | 4.923s | 9.877s | 19.973s |
| PuDDing | 0.069s | 0.134s | 0.260s | 3.946s | 7.926s | 16.039s |
| →Router | +0.005s | +0.005s | +0.005s | +0.005s | +0.005s | +0.005s |
| Speedup | 1.19× | 1.23× | 1.22× | 1.25× | 1.25× | 1.25× |

difficulty by training a router from the data.

Regarding the out-of-distribution generalization, we observe that PuDDing also works well on unseen dataset (BoolQ). PuDDing outperforms all other baselines except for FLAP, which works extraordinarily well on this specific dataset.

In Table 3, we provide comparisons on other language models: Vicuna and OPT. We confirm that our algorithm works better than other baselines under this setup as well.

### 6.3. LoRA Fine-tuning

Next, we compare the performance where we assume that we can recover the accuracies using LoRA (Hu et al., 2022). For PuDDing, we generate LoRA updates for each omission set (thus total 10 for these experiments). This requires additional storage space for storing 10 separate copies of LoRA weights for each omission set. However, this increase only incurs ∼2.5% increase in the total storage space. For training LoRA weights, we have followed the setup and hyperparameters used for LoRA training in shortened LLaMA (Kim et al., 2024); we have used Alpaca dataset (Taori et al., 2023) for training, as in the paper.

Table 4 provides LoRA fine-tuned results of depth pruning algorithms on zero-shot commonsense reasoning tasks, for LLaMA-3.1-8B pruned to 20% sparsity. We observe that PuDDing continues to achieve the best performance among all options evaluated. That is, the advantages of prompt-adaptivity also exists after fine-tuning.

### 6.4. More Complicated Tasks

In Table 5, we compare the performance of various depth pruning algorithms on more complicated tasks, including OpenBookQA (Mihaylov et al., 2018), MathQA (Amini et al., 2019), and MMLU (Hendrycks et al., 2021). From the results, we observe that PuDDing continues to perform better than the baselines, even though these tasks have not been observed during the training of the router.

## 7. Analysis

We now provide further analyses on PuDDing. In particular, we provide the following analyses: Wall clock speedup (Section 7.1), and visualization of omission sets for tasks (Section 7.2). In Appendix B, we conduct ablation studies.

### 7.1. Wall-clock Speedup

We now provide wall-clock analyses and estimates on the latency and throughput of the PuDDing-compressed models.

**Inference.** Table 6 presents the average wall-clock inference time comparison between the dense and PuDDing-pruned version of the LLaMA 3.1 8B, evaluated on NVIDIA A100 and RTX 6000 Ada. For PuDDing, we have pruned seven

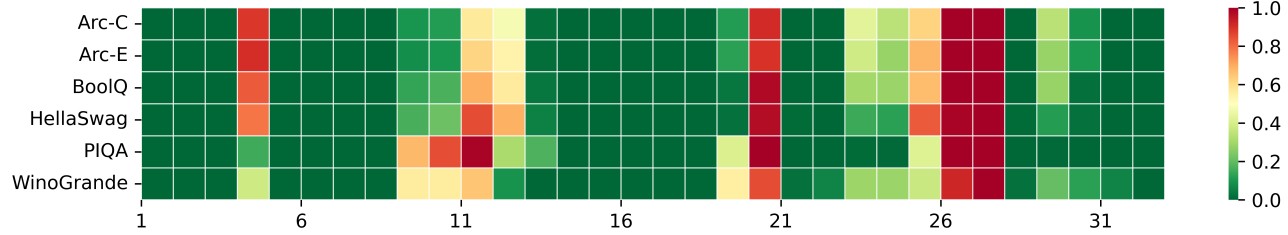

*Figure 4.* A visual illustration of the PuDDing's pruning rate of each transformer block, given the prompts drawn from various zero-shot tasks. The results are for the LLaMA 3.1 8B model, pruned to 20% sparsity (seven blocks removed). The color red indicates that the blocks are likely to be pruned, and the color green indicates that the blocks are likely to be retained. We provide additional visualizations on the other language models (OPT 6.7B and Vicuna 1.5 7B) in the Appendix C.

*Table 7.* Wall clock inference speed of the PuDDing-compressed LLaMA-3.1 8B evaluated on edge devices (M3 pro, Apple).

| M3 Pro (Apple) | Pre-fill (TTFT) | | | Pre-fill + Generation | | |
|---|---|---|---|---|---|---|
| Prompt Length | 128 | 256 | 512 | 128 | 128 | 128 |
| Gen. Length | 1 | 1 | 1 | 128 | 256 | 512 |
| Dense | 0.177s | 0.300s | 0.480s | 7.890s | 15.970s | 32.520s |
| PuDDing | 0.138s | 0.235s | 0.376s | 6.174s | 12.497s | 25.447s |
| → Router | 0.009s | 0.016s | 0.029s | 0.009s | 0.009s | 0.009s |
| Speed Up | 1.20× | 1.20× | 1.19× | 1.28× | 1.28× | 1.28× |

*Table 8.* The estimated time required to transfer the weight parameters of LLaMA-3.1 8B and PuDDing (with seven blocks pruned) to NVIDIA A100 GPU through various communication channels.

| | Bandwidth | Dense | PuDDing |
|---|---|---|---|
| PCIe Gen4 x4 | 64GB/s | 0.250s | 0.198s |
| NVIDIA NVlink | 600GB/s | 0.027s | 0.021s |

layers (21.88% sparsity). We observe that PuDDing provides a consistent 1.19-1.23× speedup during the pre-fill stage, and 1.22-1.25× speedup including the generation stage. The total routing time takes up to 4-8ms, which can be deemed negligible comparing with the overall latency. Also, Table 7 presents results on edge devices (e.g., Apple M3 Pro), showing consistent speedup. This outcome shows that the proposed method is well-suited for both server-like and edge-like hardwares.

**Parameter loading.** Table 8 presents the estimated time required for loading the model parameters of LLaMA-3.1 8B (16GB in FP32) from the storage to the GPU. PuDDing can save around 52ms on PCIe and 6ms on NVLink, which is nonnegligibly large comparing with the computational scale of running these models. However, a pitfall is that, for repeated inference, PuDDing may require loading additional weights to account for different prompts. This additional cost can be minimized by loading only the previously un-loaded blocks from the storage; in fact, many blocks overlap, as we will demonstrate in Section 7.2.

### 7.2. Pruned Block vs. Task

Figure 4 depicts the distribution of the pruned transformer blocks in LLaMA-3.1-8B model, given the prompts from different tasks. Again, we consider the case where we drop seven transformer blocks for each prompt.

From the figure, we make two intriguing observations: First, several blocks are considered almost universally unnecessary. In particular, the blocks 20, 26, 27 are removed with over 80% probability in all tasks. Similarly, there are certain block which are almost never pruned, *e.g.*, blocks 1–3 and 5–8. Second, regarding some blocks, the importance of the block highly varies over task. For instance, transformer block 4 is pruned with over 80% for ARC-Easy and ARC-Challenge. On the other hand, for PIQA and WinoGrande, the pruning rate is less than 40%; in these tasks, the blocks 9 and 10 are likelier to be less important.

We note that similar patterns can be observed for OPT and Vicuna; see Appendix C for visualizations on these models.

## 8. Conclusion

In this paper, we have developed a new paradigm for the depth pruning of large language models, where we dynamically determine which blocks should be utilized for processing the prompt given from the user. By doing so, we can save both the memory access cost and the inference computation, thus suitable for on-device deployment of large language models. We have proposed PuDDing, an algorithm to train a router using various task data. Through our experiments, we have confirmed that such framework is quite effective, clearly outperforming existing static depth pruning algorithms consistently over multiple LLMs.

**Limitations and future work.** A notable limitation of the proposed method is that we assume that we have access to various task datasets. In particular, we have focused on the case where we use LLMs for commonsense reasoning tasks, instead of an open-ended language generation. A promising future direction will be to develop new techniques

to harness unlabeled text corpus, such as Alpaca or C4, to generate diverse clusters of calibration data for attaining corresponding omission sets.

Another limitation is a general lack of mechanisms to account for the different difficulties of the tasks. For some tasks, it may be necessary to utilize all layers to generate an answer with sufficiently high quality; on the other hand, some tasks can be simply handled with very few layers. While our decision to consider a fixed number of transformer blocks is motivated by the practical constraints of on-device inference, we believe that utilizing variable-depth can be even more effective whenever the on-device memory is spacious but can be preempted to other processes.

## Acknowledgements

This work has been supported by National Research Foundation of Korea (NRF) grants funded by the Korea government (MSIT) (Nos. RS-2023-00213710, RS-2024-00453301).

## Impact Statement

Our paper targets for advancing the general field of machine learning and LLM compression. There are many potential societal consequences of our work, none which we feel must be specifically highlighted here.

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

# A. Pre-processing for the WinoGrande Dataset

The WinoGrande dataset, originally consisting of fill-in-the-blank sentences, was initially computed using the *sentence-level likelihood (sl)* as follows:

$$\text{sl}(\mathbf{z}; \mathbf{W}) = -\frac{1}{T} \sum_{i=1}^{T} \log p_i(z_i | z_{<i}; \mathbf{W}). \tag{13}$$

By reformulating the dataset into a Question-Answer format and evaluating the *task likelihood (tl)* score for the answer part using Equation (9), performance improved significantly, from 61.09% to 64.16%.

# B. Ablation Studies

We have conducted various ablation studies on the proposed algorithm, PuDDing. Below, we provide a summary of our key findings, with corresponding pointers to the relevant section.

- *Number of candidate omission sets* (Appendix B.1): We have varied the number of candidate omission sets inside the set $\mathcal{B}$, and find that having 10 classes is sufficient for handling zero-shot tasks; the gain from adding omission sets quickly saturates.
- *Proposed task likelihood score* (Appendix B.2): We compare the performance of the task likelihood-based routing and the perplexity-based routing under both static and dynamic setups. We find that the using the task likelihood score leads to a clear advantage in both scenarios.
- *MSE loss for training* (Appendix B.3): We have used the mean-squared error (MSE) loss to train the router using the soft labels. Our experiments show that this leads to a slightly better performance than using the classification loss, namely the cross-entropy loss.

## B.1. Number of Candidate Omission Sets

*Table 9.* Results of zero-shot task accuracy with varying omission set sizes.

| Number of Omission Sets | Per-task Accuracies | | | | | | Average Acc. (%) |
|---|---|---|---|---|---|---|---|
| | Arc-C | Arc-E | BoolQ | HellaSwag | PIQA | WinoGrande | |
| 5 | 40.27 | 67.85 | 61.01 | 62.60 | 73.83 | 61.56 | 61.19 |
| 10 | 41.47 | 67.09 | 62.02 | 62.92 | 73.94 | 64.16 | **61.93** |
| 30 | 38.57 | 66.88 | 63.70 | 64.23 | 72.85 | 63.93 | 61.69 |

Table 9 shows the accuracy of zero-shot task with varying omission set sizes, comparing the impact of using 5, 10, and 30 omission sets across common-sense reasoning tasks.

Using only 5 omission sets results in a lower accuracy, with an average of 61.19%, as it shows insufficient performance for optimal results. 30-set configuration, despite some task-specific advantages, does not lead to consistently higher performance. In contrast, the 10-set configuration provides an improvement across multiple tasks, with a highest average accuracy of 61.93%, indicating that this size offers a better balance between performance and model efficiency.

## B.2. Effectiveness of the Proposed Task Likelihood Score

*Table 10.* Zero-shot accuracy performance of static pruning methods.

| Method | Metric | Task-wise | Average Acc. (%) |
|---|---|---|---|
| SLEB | Batch-ppl | ✗ | 57.24 |
| | Batch-ppl | ✓ | 59.32 |
| Static PuDDing (Ours) | task likelihood (tl) | ✓ | **61.02** |

*Table 11.* Zero-shot accuracy performance of dynamic routing pruning methods.

| Method | Metric | Omission Set Selection | Router Training | Average Acc. (%) |
|---|---|---|---|---|
| SLEB | Prompt-ppl | per-prompt dynamic | ✗ | 52.58 |
| | Batch-ppl | pre-selected 10 sets | ✓ | 58.19 |
| PuDDing (Ours) | task likelihood (tl) | pre-selected 10 sets | ✓ | **61.93** |

Both dynamic and static pruning methods were set up on LLaMA-3.1 8B, with a sparsity of 20%, corresponding to the pruning of seven blocks.

Static pruning experiments in Table 10 show the experimental validation of task-adaptive pruning and our proposed *tl* scoring method. First, we compare two pruning strategies using batch-PPL: one that applies a fixed omission set based on a Wikitext-calibrated batch (57.24% accuracy) and another that dynamically selects omission sets per task (59.32% accuracy). The improvement confirms the claim that different tasks require different layer sets. Next, keeping the task-wise adaptive setting constant, we replace batch-PPL with our *task likelihood (tl)* loss for omission set selection. This further improves accuracy from 59.32% to 61.02%, demonstrating that our method is more effective at identifying layers that impact quality of task-specific inference.

By comparing Batch-PPL Router (58.19%) with PuDDing (Task Likelihood Loss) (61.93%) in Table 11, we observe that our *tl* metric results in omission sets that generalize better across tasks, contributing to an additional performance gain. When comparing these Table 11 results with Table 10, Batch-PPL static pruning (57.24%) improves with router training (58.19%), and our method (60.12%) also benefits from router training, increasing to 61.93%. This validates that even within the same task, different prompts favor slightly different omission sets, and adapting omission sets dynamically through router training is essential for optimal pruning.

## B.3. Using MSE Loss for Training

*Table 12.* Zero-shot accuracy comparison between three different training strategy on the router. The best performances are marked in **bold**, and the runner-up is marked with underline.

| Label | Loss | Per-task Accuracies | | | | | | Average Acc. (%) |
|---|---|---|---|---|---|---|---|---|
| | | Arc-C | Arc-E | BoolQ | HellaSwag | PIQA | WinoGrande | |
| One-hot Vector | CE | 41.21 | 66.29 | 59.66 | 61.44 | 72.96 | 59.35 | 60.15 |
| Log-likelihood | CE | 39.67 | 67.80 | 61.77 | 60.80 | 73.56 | 59.04 | 60.44 |
| Log-likelihood | MSE | 41.47 | 67.09 | 62.02 | 62.92 | 73.94 | 64.16 | **61.93** |

In Table 12, we present the result of zero-shot task accuracies in the different router training settings. For the label, a one-hot vector signifies that the router learns only from the highest confidence value within the omission block set derived from the training dataset. In contrast, the log-likelihood label allows the router to incorporate all confidence values during training. Our findings show that training with log-likelihood label leads to improved average accuracy (from 60.16% to 61.93%). Hence, we observe that the mean-squared error (MSE) loss function outperforms cross entropy (CE). As a result, in this case for routers to train for finding optimal omission sets by given prompts, richer information in the label (i.e., un-chosen labels are not assigned to zero values), and employing MSE loss enhances better performance.

## B.4. Training the Router with Multi-Domain Calibration Datasets

*Table 13.* Zero-shot accuracy comparison of dense, PuDDing, and PuDDing-MultiDomain using diverse calibration datasets.

| Method | Average Acc. (%) | Per-task Accuracies | | | | | | | | |
|---|---|---|---|---|---|---|---|---|---|---|
| | | Arc-C | Arc-E | BoolQ | HellaSwag | PIQA | WinoGrande | MathQA | PubMedQA | SciQ |
| Dense | 73.42 | 53.50 | 81.52 | 82.20 | 78.81 | 79.98 | 73.40 | 39.53 | 75.80 | 96.00 |
| PuDDing | 61.28 | 41.47 | 67.09 | 62.02 | 62.92 | 73.94 | 64.16 | 27.27 | 60.00 | 92.70 |
| PuDDing-MultiDomain | 62.37 | 41.38 | 67.26 | 67.37 | 63.68 | 73.07 | 64.56 | 29.58 | 62.00 | 92.40 |

In Section 6, we construct omission sets using common-sense reasoning datasets that are widely used to evaluate language models' reasoning ability—ARC, PIQA, HellaSwag, and WinoGrande—to reflect popular usage scenarios. For the new variant, PuDDing-MultiDomain, we conduct an additional experiment by incorporating calibration datasets from diverse domains: MathQA (mathematics), PubMedQA (biomedical), and SciQ (science). To maintain consistency in sample size with the original setting, we include only a subset of the original common-sense datasets—specifically, ARC-Easy and WinoGrande—in this experiment. All other experimental settings follow the same setup described in Section 6.1.

We observe that PuDDing-MultiDomain achieves slightly improved average accuracy compared to the original PuDDing, with noticeable gains in the newly introduced datasets (e.g., +2.31%p in MathQA, +2.00%p in PubMedQA). This indicates that the router can benefit from more diverse calibration dataset, especially when deployed in tasks requiring domain-specific knowledge. Overall, our method demonstrates consistent performance across both in-domain and out-of-domain tasks, and

can be further customized to specific user applications by adjusting the coverage of the omission set pool.

## B.5. Quantized PuDDing

*Table 14.* combined with other compression techniques

| LLaMA-3.1-8B | Average | Arc-C | Arc-E | BoolQ | HellaSwag | PIQA | WinoGrande |
|---|---|---|---|---|---|---|---|
| Dense | 74.90 | 53.50 | 81.52 | 82.20 | 78.81 | 79.98 | 73.40 |
| SLEB | 57.24 | 34.90 | 66.25 | 49.11 | 61.60 | 74.37 | 57.22 |
| Shortened LLaMA | 55.77 | 34.30 | 65.15 | 44.52 | 60.55 | 73.67 | 56.43 |
| PuDDing | 61.93 | 41.47 | 67.09 | 62.02 | 62.92 | 73.94 | 64.16 |
| PuDDing + W8A16 (AWQ) | 61.68 | 41.30 | 67.00 | 61.50 | 62.95 | 73.72 | 63.61 |
| PuDDing + W4A16 (AWQ) | 58.58 | 37.37 | 61.45 | 60.64 | 57.55 | 71.71 | 62.75 |

In Table 14 presents the performance of the proposed method when combined with a representative compression technique such as quantization (e.g. Activation-aware Weight Quantization (AWQ) (Lin et al., 2024)). Interestingly, we observe that compressing the weights to 8-bit results in no performance degradation. Although 4-bit quantization introduces some degradation, the performance still surpasses that of static pruning methods applied with BF16 precision.

## C. Additional Visualizations

Figure 5 illustrates the dynamic block selection process in various tasks, highlighting that this process has also been analyzed with different models to highlight how the block selection strategy varies not only varying tasks but also depending on the specific architectures of the models.

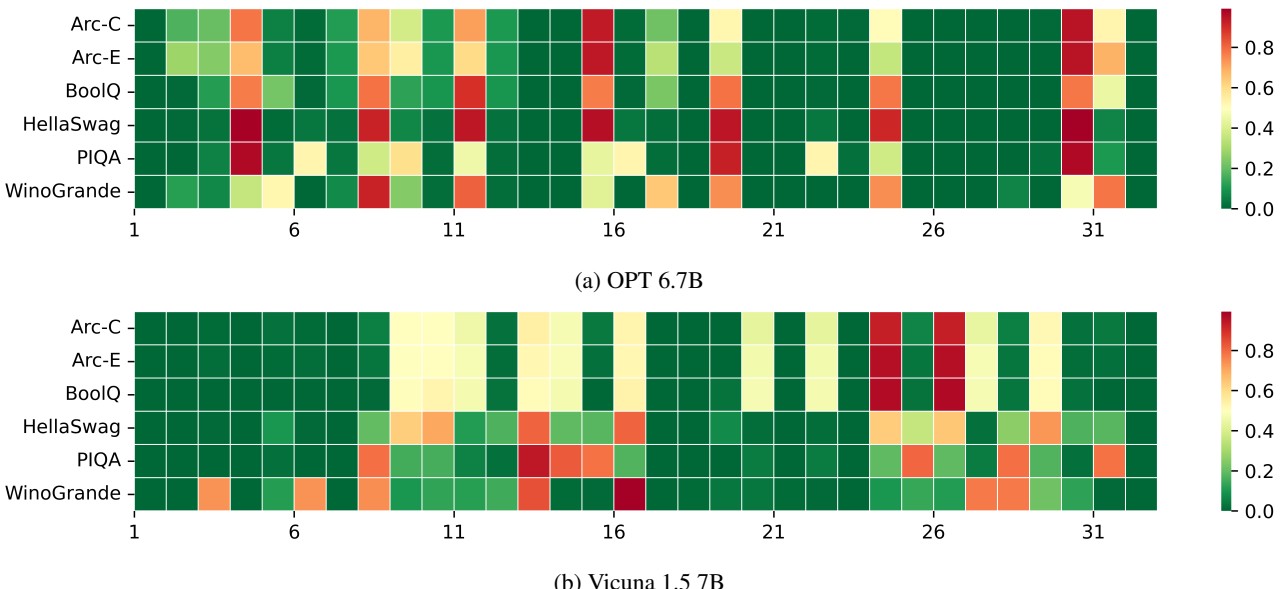

(a) OPT 6.7B

(b) Vicuna 1.5 7B

*Figure 5.* A visual illustration of the PuDDing's pruning rate of each transformer block, given the prompts drawn from various zero-shot tasks. The results are for the OPT 6.7B model and vicuna 1.5 7B, pruned to 20% sparsity (seven blocks removed). The color red indicates that the blocks are likely to be pruned, and the color green indicates that the blocks are likely to be retained.

