# OpenReview forum: "Prompt-based Depth Pruning of Large Language Models"
_ICML.cc/2025/Conference — ICML 2025 poster_

### Official Review · Reviewer_9fAo · 2025-03-08

**Overall Recommendation:** 3

**Summary:**

This paper propose prompt-based depth pruning of Large Language Models, where given a prompt, a router is trained to select a best set of LLMs layers, and other layers will be pruned.

**Claims And Evidence:**

Yes

**Essential References Not Discussed:**

No

**Experimental Designs Or Analyses:**

No issues

**Methods And Evaluation Criteria:**

Yes

**Other Comments Or Suggestions:**

NA

**Other Strengths And Weaknesses:**

Strength: This paper first show supportive observations that the effectiveness of  LLMs internal layers are prompt dependent, which give empirical evidence that some layers can be disregard and abandoned for a specific prompt.

Weakness: It would be better to show the training convergence analysis of the router

**Questions For Authors:**

NA

**Relation To Broader Scientific Literature:**

This work contributes to efficient LLM deployment.

**Theoretical Claims:**

NA

---

> ### Author Rebuttal · Authors · 2025-03-31
>
> Dear reviewer 9fAo,
>
> We appreciate your positive evaluation on our paper, acknowledging the practical advantages of our method. We respond to the concerns you raise in what follows.
>
> ---
>
> ### **It would be better to show the training convergence analysis of the router.**
>
> Following your suggestion, we have attached the loss curves of the router training in the following [LINK (training loss)](https://ibb.co/VYsX2gYf ), and [LINK (test loss)](https://ibb.co/5XkTWZpq). The results confirm that the test loss of the router has successfully converged by the end of the training.
>
> We will add the plot in the revised manuscript.
>
> ---
>
> We hope that you find our response reasonable. Please let us know if you have any further questions.
>
>
> Best regards,
> Authors.

---

> > ### Comment · Reviewer_9fAo · 2025-04-04
> >
> > Thanks for your responsible response. I maintain my positive score.

---

> > > ### Author Response · Authors · 2025-04-04
> > >
> > > Dear reviewer 9fAo,
> > >
> > > Thank you for your kind attention to our response. We appreciate your time and consideration.
> > > Please do not hesitate to let us know if you have any further questions.
> > >
> > > Best regards,
> > > Authors

---

### Official Review · Reviewer_KB85 · 2025-03-11

**Overall Recommendation:** 3

**Summary:**

This paper propose a input-dependent depth pruning method for LLMs. Unlike existing static model pruning methods where the LLM is pruned into a small subnetwork and apply it for all testing examples, this paper considers select differently pruned sub-networks for inference depends on the task that the testing example belongs to.

The algorithm is very straightforward and contains the following parts: 1. depth-pruned network candidates ("candidate omission set generation" called in thiss paper). The authors first collect a series of sub-networks by perform depth-pruning of the LLMs on a set of different datasets. During testing, a router will select one of these sub-networks (each represented as a set of indices of the pruned layers) for each input example. 2. router training. Selecting sub-networks is formulated into a regression problem followed by argmax operation. The authors will pair each example in the training dataset with the loss values given each sub-network. The router is trained to predict these loss values given the input example. 3. Inference. During inference, the router first predict the losses of each sub-network, then the sub-network with the minimum predicted loss is selected for inference.

The author demonstrate the proposed method outperforms other depth pruning methods. By pruning ~20% parameters, the proposed method brings 13% performance degradation.

**Claims And Evidence:**

The claims made in the submission is well supported by empirical results. The authors first present an empirical test regarding the importances of transformer blocks given different inputs in Figure 2. This verify the correctness of this paper's focus: the model parameters could be input-dependent, and we can only select the necessary parameters (layers) for different inputs.

**Essential References Not Discussed:**

Are there related works that are essential to understanding the (context for) key contributions of the paper, but are not currently cited/discussed in the paper? Be specific in terms of prior related findings/results/ideas/etc.

The key contribution of this paper is a dynamic pruning method. However, a very similar paper is missing. Also, although the authors formulate the story of this paper as "model pruning", this paper actually closely aligns with the research of **contextual sparsity**.

1. Missing paper on dynamic model pruning [1]. This paper also focuses on dynamic pruning and select the sub-network given different inputs. Furthermore, the authors in [1] demonstrate their dynamic pruning methods only brings less than 2% performance degradation on the same datasets evaluated in this paper's experiments while achieves up to 75% sparsity. This paper only achieves 20% sparsity while hurting the performance by more than 10%.
2. Missing discussion on **contextual sparsity**. This paper's "dynamic pruning" can be interpreted as contextual sparisty, whether the sub-network is selected based on the input during inference. Typical contextual sparsity methods [2-5] also do not fine-tune the LLM. They collect the parameter activation patterns on training dataset and train a router to predict these activated parameters, which is very similar to the proposed method in this paper.

It is necessary to include these papers into the related work section and discuss the differences.

[1] Hou, Bairu, et al. "Instruction-Following Pruning for Large Language Models." *arXiv preprint arXiv:2501.02086* (2025).

[2] Liu, Zichang, et al. "Deja vu: Contextual sparsity for efficient llms at inference time." International Conference on Machine Learning. PMLR, 2023.

[3] Akhauri, Yash, et al. "Shadowllm: Predictor-based contextual sparsity for large language models." arXiv preprint arXiv:2406.16635 (2024).

[4] Zhou, Yang, et al. "Sirius: Contextual sparsity with correction for efficient llms." arXiv preprint arXiv:2409.03856 (2024).

[5] Lee, Donghyun, et al. "Cats: Contextually-aware thresholding for sparsity in large language models." arXiv preprint arXiv:2404.08763 (2024).

**Experimental Designs Or Analyses:**

The experiment design is reasonable. I do not find significant issues.

**Methods And Evaluation Criteria:**

The method is generally reasonable and straightforward. The evaluation datasets are also proper. I only have several questions regarding on the method design, where I think the following methods could be better than the proposed method:

- Method design 1: Why not introduce a learnable network that predicts a mask $m_i\in {0,1}$ for each layer $i$ ? This is a typial method for structured model pruning. You can directly train the parameters of this learnable network while fixing the model to be pruned by optimization the following objective function:

$$\ell = \mathcal{L}_{\text{CE}}[f(\boldsymbol x; \boldsymbol m \cdot \boldsymbol\theta), \boldsymbol y]$$,

where $\boldsymbol m = g(x, \boldsymbol \phi)$ is the output of the learnable network with parameter $\boldsymbol \phi$, $m\cdot \theta$ means we mask the layers of the LLM's parameter $\theta$ if $m_i = 0$ (otherwise we keep it). This is more straightforward and allows you to select sub-networks with more freedom, instead of only have 10 different subnetworks for selection.

- Method design 2: why not directly train a classifier? The authors propose to train the router with the regression objective and predict the loss values with each sub-network. I wonder the rationale behind such a design.

**Other Comments Or Suggestions:**

N/A

**Other Strengths And Weaknesses:**

Please refer to the **Questions For Authors**. The main weaknesses include the poor performance, missing references, similarity to contextual sparisty methods, and method design.

**Questions For Authors:**

1. The discussion with missing relatedd work (see **Essential References Not Discussed**)
2. The performance degradation is still signficiant. From my perspective, it is easy to propose a method that brings a better tradeoff between sparisty and performance than baselines, but it is hard (and more important) to propose a method that brings a **usable** tradeoff. The current method occurs more than 10% performance degradation while achieves 20% sparsity, which is far away from usable for modern LLMs. The poor performance (although still better than baselines) makes this paper less competitive.
3. Why do not adopt the method design 1 (see **Methods And Evaluation Criteria**) which is very straightforward and familiar to people in the research area of model pruning?  I think that method could bring better performance given the more freedom and expressivity. It is also a commonly used method for structured model pruning [1].

[1] Xia, Mengzhou, et al. "Sheared llama: Accelerating language model pre-training via structured pruning." *arXiv preprint arXiv:2310.06694* (2023).

**Relation To Broader Scientific Literature:**

N/A

**Theoretical Claims:**

N/A. This paper is mainly about empirical findings.

---

> ### Author Rebuttal · Authors · 2025-03-31
>
> Dear reviewer KB85,
> Thank you for your constructive and thoughtful feedback.  In what follows, we address the concerns you raised one-by-one.
>
> ---
>
> ### **Missing paper on dynamic model pruning [1]**.
>
> Thank you for the pointer to this concurrent work. The paper [1] missed our attention, as it appeared on arXiv after finalizing our paper (Jan 2025). Still, it looks interesting and relevant; we will cite and discuss it in the revised manuscript.
>
> We note that [1] differs from our work in several key aspects, making two approaches complementary to each other.
> - **Sparsity structure:** Our work considers depth pruning (very coarse) which is friendlier to hardware using grouped computations (c.f. Kim et al. (2024)), On the other hand, [1] considers row/col pruning.
> - **Training footprint:** Our work only trains the router and keeps the base LLM untouched (except for Section 6.3). On the other hand, [1] involves training the base LLM much further, conducting both pre-training and fine-tuning.
> Overall, these differences make two approaches complementary to each other---ours focusing on versatility and [1] focusing on good sparsity-quality tradeoff---making both valuable contributions with distinct scopes.
>
> Kim et al., “Shortened LLaMa: A simple depth pruning for large language models,” arXiv 2024.
>
> ---
>
> ### **This paper only achieves 20% sparsity while hurting the performance by more than 10%. (…) The performance degradation is still significant, comparing with dynamic pruning.**
>
> As we have explained briefly above, our work imposes (1) coarse sparsity structure, and (2) minimal retraining, which increases the performance degradation given the same sparsity. This is because we intend to make the algorithm easily adoptable by low-budget end users, who might be the ones that want to develop their own input-dependent depth-pruned LLMs specialized for their tasks. In this sense, we are trading the performance for better usability.
>
> ---
>
> ### **Missing discussion on contextual sparsity**.
>
> Thank you for pointing this out. The reviewer is correct; we will add discussions on [3,4,5] (note that we already discuss [2] in Section 2).
>
> ---
>
> ### **Method design 1: Why not introduce a learnable network that predicts a mask for each layer?**
>
> We have decided to confine the routing decisions to pre-determined options to make the routing decision easier and simpler (only 10 choices, much less than $2^{32}$), so that we can train a lightweight router with limited amount of training data. In fact, we have also tried a learnable mask approach, but have failed to train a sufficiently light yet performant router. Such difficulty is circumvented in existing works by using the intermediate features to make routing decisions (as in D-LLM) or joint training with base LLM parameters (as in Hou et al. (2025)). We have deliberately avoided these options to enable a one-time memory load and enhance the usability of our technique by GPU-poor end users, respectively.
>
> ---
>
> ### **Method design 2: Why not directly train a classifier?**
>
> By conducting a regression on the likelihoods, we are basically training using the soft label instead of hard labels. Training with such objectives is known to enjoy better generalization performance, by learning to mimic the “dark knowledge” in these models. In fact, this is a popular technique in knowledge distillation; Hinton et al. (2015) shows that one can approximate knowledge distillation by the $ell_2$ regression of pre-softmax activations.
>
> Empirically, in Table 11, we have compared the regression-trained router against the classification-trained router, where we observe that the regression-trained router performs better.
>
> Hinton et al. “Distilling the knowledge in a neural network,” arXiv 2015
>
> ---
>
> We sincerely hope that you find our response reasonable. Please don’t hesitate to let us know if there are any further questions.
>
> Best regards,
> Authors

---

> > ### Comment · Reviewer_KB85 · 2025-04-04
> >
> > I thank the authors for their detailed response. The paper and the rebuttal are reasonable to me. I will keep my positive rating and learn to accept this paper.

---

> > > ### Author Response · Authors · 2025-04-05
> > >
> > > Dear Reviewer KB85,
> > >
> > > We sincerely appreciate your positive feedback and helpful suggestions. We're glad that our responses have been satisfactory and will incorporate the additional discussions from our rebuttal into the final version.
> > >
> > > If you believe your concerns have been adequately addressed, we would be grateful if you could consider raising the score. If there are any remaining issues, we would be more than happy to address them during the remaining period.
> > >
> > > Best regards,
> > > The Authors

---

### Official Review · Reviewer_r1RD · 2025-03-13

**Overall Recommendation:** 3

**Summary:**

This paper introduces Prompt-routed Dynamic Depth Pruning (PuDDing), a method for dynamically pruning transformer blocks in LLMs based on the input prompt. The core motivation is that the importance of transformer layers is task-dependent, making static pruning suboptimal. To address this, PuDDing trains a lightweight router that predicts the optimal omission set for each prompt, reducing inference costs while maintaining accuracy.

The proposed approach consists of two key steps:

1) Candidate Omission Set Generation: A small, diverse set of omission strategies is precomputed using a new task likelihood (TL) loss, which improves on traditional perplexity-based pruning metrics.
2) Router Training: A BERT-based classifier is trained to predict the best omission set for a given prompt, ensuring minimal accuracy loss while improving efficiency.

Empirical results demonstrate that PuDDing outperforms static depth pruning methods (e.g., SLEB, Shortened LLaMA) by up to 4% in zero-shot accuracy on commonsense reasoning tasks (ARC, BoolQ, WinoGrande). It also achieves a 1.2× inference speedup compared to the unpruned model. The method is particularly designed for on-device inference, as it loads only the necessary transformer layers from storage, reducing memory requirements.

Overall, the paper contributes to model compression and efficient inference research by demonstrating that depth pruning can be made adaptive to input prompts, improving efficiency without requiring extensive retraining or hardware-specific optimizations.

## Update after rebuttal
The authors have done a good job by adding experiments and analysis during the rebuttal which strengthens the paper but I still believe that the idea is not very novel and might need more ingenuity to increase it's impact. Also, the gains are marginal thereby making this technique not useable on it's own. I would suggest the authors to come up with a truly dynamic approach which might help in improving the speedup gains. Good Luck to the authors

**Claims And Evidence:**

Below is an evaluation of key claims:

1) Claim: Transformer block importance is task-dependent.
- Evidence: Section 3 presents empirical results showing that pruning different blocks affects different tasks differently.
- Support: Figure 2 demonstrates how removing specific layers leads to varying accuracy drops across datasets.
- Weakness: While the empirical evidence is strong, the paper lacks theoretical justification for why this happens at a structural level.

2) Claim: PuDDing achieves inference speedup.
- Evidence: Table 6 shows a 1.2x speedup in inference over unpruned models.
- Support: The method reduces the number of active layers, which naturally leads to speed improvements.
- Weakness: The speedup is modest compared to other pruning methods (e.g., structured sparsity, quantization).

3) Claim: PuDDing is a fully dynamic pruning method.
- Issue: The method selects from precomputed omission sets rather than dynamically pruning layers on a per-query basis. A true dynamic pruning method would adjust pruning layer-by-layer rather than choosing from a fixed set.

4) Claim: Task likelihood (TL) loss is superior to perplexity for pruning.
- Issue: The paper provides empirical comparisons (Table 9, Table 10) but does not explain why TL loss is theoretically better. A mathematical argument or formal connection to task complexity would strengthen this claim.

**Essential References Not Discussed:**

Some of the prior work that author could have included:

- Mixture-of-Depths (MoD) (Raposo et al., 2024) introduces adaptive layer skipping, where transformer blocks are selectively skipped based on token-level routing decisions.
- D-LLM (Wang et al., 2024) also uses a router for adaptive pruning, but dynamically decides layer usage per token.

MoD and D-LLM perform token-level routing (more flexible, but computationally expensive). PuDDing precomputes a small set of pruning strategies and selects one per prompt (faster but less flexible).

**Experimental Designs Or Analyses:**

The experimental design is mostly sound. Below is an analysis of the strengths and few weaknesses of the experiments.

1) Comprehensive Benchmarking on Commonsense Reasoning Tasks
- The paper evaluates PuDDing on six widely used benchmarks (ARC, BoolQ, PIQA, WinoGrande, HellaSwag).
- Multiple pruning baselines (SLEB, Shortened LLaMA, FLAP, SliceGPT) are included for fair comparison.
- Accuracy is measured consistently across different pruning levels (10%, 15%, 20%), ensuring robustness.

2) Fair Comparisons with Static Pruning Methods
- The experimental setup controls for model size and number of pruned layers, ensuring a fair comparison between static and dynamic pruning.
- The study includes ablation experiments on router training methods (Table 11) and omission set selection (Table 8), adding depth to the analysis.

3) Inference Speed Evaluation
- Table 6 reports wall-clock speed improvements, showing a 1.2× speedup over unpruned models.
- The memory efficiency claim is supported by parameter transfer time comparisons (Table 7).

4) No Real-World Deployment Results
- The paper claims PuDDing is well-suited for on-device inference, but all experiments are conducted on high-end GPUs (A100, RTX 6000).
- Missing mobile/edge device benchmarks (e.g., ARM-based chips, Jetson devices) weakens the practical relevance of the method.

**Methods And Evaluation Criteria:**

1) Relevant Benchmark Datasets for Task-Specific Evaluation
- The paper evaluates PuDDing on six widely used commonsense reasoning datasets (ARC, BoolQ, PIQA, WinoGrande, HellaSwag).
- Since the goal is to show that different tasks require different pruning strategies, these benchmarks make sense.

2) Fair Comparisons Against Strong Baselines
- The method is compared against multiple pruning techniques, including: Static Depth Pruning (SLEB, Shortened LLaMA)
Width Pruning (FLAP, SliceGPT)
- Multiple sparsity levels (10%, 15%, 20%) are tested, ensuring fairness in evaluation.

3) Ablation Studies for Router Training
- The paper evaluates different loss functions (MSE vs. CE) for the router, showing that MSE improves generalization (Table 11).
- The number of candidate omission sets is also analyzed (Table 8).

4) No Real-World Deployment Tests
- The paper claims PuDDing is suitable for on-device inference, but does not test it on resource-limited devices.
- All experiments are done on NVIDIA A100 / RTX 6000 GPUs, which do not reflect real-world constraints of mobile or edge hardware.

**Other Comments Or Suggestions:**

The paper is well-organized and well-written. There are minor typos:
"langauge" --> "language" (Section 5)
"calibraion” --> “calibration" (multiple instances)

**Other Strengths And Weaknesses:**

Strengths

1) Practical and Computationally Efficient Approach
- PuDDing introduces a lightweight routing mechanism that selects omission sets once per prompt, reducing computational cost compared to token-level routing methods like MoD or D-LLM.
- Unlike fine-grained dynamic pruning, PuDDing does not require per-token decisions, making it more efficient for real-time inference.

2) Strong Empirical Performance on Task-Specific Pruning
- The experiments show that task-dependent pruning improves accuracy over static pruning methods by up to 4% in zero-shot commonsense reasoning tasks.
- This reinforces the idea that layer importance varies by task, a key insight for adaptive pruning.

3) Hardware-Agnostic Design
- Unlike width pruning (which often requires hardware-specific optimizations), PuDDing’s depth pruning approach can be used on any hardware without requiring changes to matrix sparsity patterns.

4) Good Paper Organization and Clarity
- The paper is well-structured, with clear motivation, experimental design, and results.
- The figures (e.g., Figure 2 on transformer block importance) effectively communicate key insights.

Weaknesses

1) Limited Novelty – More of an Incremental Contribution
- While the idea of prompt-aware pruning is new, it builds upon static pruning (SLEB) and dynamic routing (Mixture-of-Depths, D-LLM) but does not introduce a fundamentally novel pruning algorithm.

2) Fixed Omission Set Reduces Flexibility
- The pruning method is not truly dynamic—it selects from a small precomputed set of omission strategies, instead of pruning layer-by-layer in real-time. A more adaptive method would dynamically select layers per query, instead of relying on precomputed omission sets.

3) No Real-World Deployment or Mobile Testing
The paper claims that PuDDing is suitable for on-device inference, but all experiments are done on high-end GPUs (A100, RTX 6000).
Missing evaluation on real constrained devices (e.g., ARM CPUs, mobile GPUs) makes this claim untested.

4) Relatively Modest Speedup (1.2×)
The reported 1.2× inference speedup is not very high compared to other compression techniques (e.g., structured sparsity, quantization, MoE models).

**Questions For Authors:**

1) How does PuDDing compare to other dynamic pruning methods like Mixture-of-Depths (MoD) or D-LLM?
2) How well does the router generalize to unseen tasks or domains?
3) The method selects a pruning strategy per prompt, but this means that different layers may need to be loaded dynamically from storage. How does the dynamic loading of omission sets affect inference latency?
4) Can PuDDing be applied beyond transformer models?

**Relation To Broader Scientific Literature:**

The paper is related to prior work in model compression, pruning techniques, and adaptive computation for LLMs.

1) Static Depth Pruning: SLEB (Song et al., 2024), Shortened LLaMA (Kim et al., 2024) perform static depth pruning, removing less important transformer blocks based on perplexity or activation-based importance metrics. PuDDing extends these methods by introducing prompt-conditioned pruning decisions, making depth pruning task-adaptive rather than fixed.

2) Width Pruning & Sparsity Methods: FLAP (An et al., 2024), SliceGPT (Ashkboos et al., 2024) apply structured width pruning, reducing parameters in weight matrices. PuDDing focuses on depth pruning instead of width pruning, making it hardware-agnostic.

**Theoretical Claims:**

The paper does not present any formal theoretical proofs. Instead, it relies entirely on empirical evidence to support its claims.

Theoretical justification cane be provided for following:
1) why TL loss is a better metric
2) how well the router generalizes to unseen tasks, given that it is trained on specific datasets

---

> ### Author Rebuttal · Authors · 2025-04-01
>
> Dear reviewer r1RD,
> Thank you for your constructive feedback. In what follows, we respond to the points raised by the reviewer one-by-one.
>
> ---
>
> ### **No Real-World Deployment Results (...) Missing mobile/edge device benchmarks weakens the practical relevance of the method.**
> Thank you for this comment. Following the reviewer’s suggestion, we have measured performance of LLaMA-3.1-8B PuDDing on MacBook Pro, where the inference took place on the M3 Pro chip (ARM-based processor) with 18GBs of RAM (we run using C++).
> ||Pre-fill (TTFT)|||Generation|||
> |:-|:-:|:-:|:-:|:-:|:-:|:-:|
> |Prompt Len|128|256|512|128|256|512|
> |Gen.Len| 1 | 1 | 1 |128|256|512|
> |Dense|0.177|0.300|0.480|7.890|15.970|32.520|
> |PuDDing|0.138|0.235|0.376|6.174|12.497|25.447|
> |Router|0.009|0.016|0.029|0.009|0.009|0.009|
> |Speedup|1.20×|1.20×|1.19×|1.28×|1.28×|1.28×|
>
> We will add this result in the revised manuscript.
>
> ---
>
> ### **How does the dynamic loading of omission sets affect inference latency?**
> As shown in Table 7, the initial loading of PuDDing takes ~0.2s using PCIe Gen4 (0.02s when using NVlink). This latency is relatively small comparing with the latency from the running the generation steps, which can take more than 16s for generating 512 tokens using RTX 6000 Ada.
> We will add the relevant discussion to the revised version.
>
> ---
>
> ### **The reported 1.2x inference speedup is not very high compared to other compression techniques.**
> First, we highlight that this 1.2x speedup is the ***real speedup,*** meaning that it can be achieved on almost all hardwares, without any implementational tricks. This is in stark contrast with many structured pruning algorithms (which often fails to provide speedup on hardwares that use grouped operations, e.g., GPUs with tensor cores) or quantization methods (which needs low-precision hardware and/or needs frequent dequantization).
> Second, the proposed PuDDing can be combined with other compression techniques, providing orthogonal benefits. For instance, we can apply weight quantization on the PuDDing to reduce the computational cost even further; W8 quantization can be done almost for free on this model, and applying W4 quantization is still better than the baselines.
> |LLaMA-3.1-8B|AVG|Arc-C|Arc-E|BoolQ|HellaSwag|PIQA|Winogrande|
> |:-|:-:|:-:|:-:|:-:|:-:|:-:|:-:|
> |Dense|74.90|53.50|81.52|82.20|78.81|79.98|73.40|
> |SLEB|57.24|34.90|66.25|49.11|61.60|74.37|57.22|
> |Shortened LLaMA|55.77|34.30|65.15|44.52|60.55|73.67|56.43|
> |PuDDing|61.93|41.47|67.09|62.02|62.92|73.94|64.16|
> |PuDDing+ w8a16 (AWQ)|61.68|41.30|67.00|61.50|62.95|73.72|63.61|
> |PuDDing + w4a16 (AWQ)|58.58|37.37|61.45|60.64|57.55|71.71|62.75|
> We will add the relevant discussion to the revised version.
>
> ---
>
> ### **The pruning method is not truly dynamic.**
> We agree with the reviewer’s point. To avoid any confusion, we will revise the manuscript to avoid calling our method “dynamic” and replace it with other options, such as “contextual.”
>
> ---
>
> ### **Fixed omission set reduces flexibility.**
> While this is true, using the fixed omission set was a necessary choice to reduce the complexity of the routing task, thereby minimizing the size of the router and its associated training costs (both compute and data). In fact, we have tried a “truly dynamic” approach similar to D-LLM, but this option worked worse than the current version.
> For a more detailed answer, please refer to our ***answer #4 to the reviewer KB85.***
>
> ---
>
> ### **Some of the prior work that author could have included (...) How does PuDDing compare to other dynamic pruning methods like MoD or D-LLM?**
> We clarify that we have made conceptual comparisons against MoD and D-LLM in the sections 1 and 2.2. The works critically differ from our work in that they require loading full models to memory to conduct token-level routing.
>
> ---
>
> ### **Why TL loss is a better metric.**
> The task likelihood loss measures the perplexity (PPL) of the sample, but measured on the “answer” part of the given sample conditioned on the “question,” i.e.,   $P(x_{\mathrm{answer}}|x_{\mathrm{question}})$.   This improves the depth pruning decisions, by letting us optimize how well we answer to the questions (of specific type), not focusing on the fluency in general.
>
> ---
>
> ### **How well does the router generalize to unseen tasks or domains.**
> We empirically confirm that it has strong generalizability to unseen and specialized tasks such as MMLU/MathQA/OBQA (Table 5), and MathQA/SciQA (newly added). Due to the character limit, we refer the reviewer to our ***answer#1 to the reviewer 2qci.***
>
> ---
>
> ### **Other comments.**
> Unfortunately, we cannot give detailed answers to all points in this round due to the space limit. We deeply appreciate these, and will incorporate these sincerely in the revised manuscript.
>
> ---
>
> We sincerely hope that you find our response reasonable. Please don’t hesitate to let us know if there are any further questions.
>
> Best regards,
> Authors

---

> > ### Comment · Reviewer_r1RD · 2025-04-04
> >
> > I thank the authors for their detailed response. I am satisfied with most of their arguments and would suggest them to include the additional results in the final version of the paper. I will maintain my positive review of this paper.

---

> > > ### Author Response · Authors · 2025-04-04
> > >
> > > Dear Reviewer **r1RD**,
> > >
> > > We appreciate the reviewer’s positive feedback and helpful suggestions. We are pleased that our responses have been satisfactory; we will include the additional results in the final version as recommended.
> > > If you feel that your concerns have been addressed well, we sincerely ask you to consider raising the score. If there are any remaining concerns, we will be very happy to address further in the remaining period.
> > >
> > > Best regards,
> > > Authors.

---

### Official Review · Reviewer_2qCi · 2025-03-14

**Overall Recommendation:** 3

**Summary:**

This paper introduces PuDDing, a method to reduce LLM inference costs by skipping transformer layers on a per-input basis. The motivation is that different tasks or queries may not require all layers of a deep model. PuDDing consists of two main components: (1) a procedure to generate a small set of candidate omission sets, and (2) a lightweight router network that, given a new prompt, predicts the best omission set to use. The authors train the router by creating a training dataset of prompts and the optimal omission decisions for those prompts, found via a data-driven search (they evaluate different layer-drop combinations on a set of training prompts to see which yields minimal loss). Once trained, the router can generalize to new prompts. On several commonsense reasoning benchmarks (ARC, PIQA, WinoGrande, BoolQ, etc.), a PuDDing-pruned LLaMA-3.1-8B model at 20% sparsity outperforms equivalent static-pruned model.

**Claims And Evidence:**

The paper claims that prompt-adaptive layer pruning yields better task performance than static pruning for the same speedup, and that it achieves a meaningful speedup over the dense model. The evidence supports these claims yet the scope of the task is too narrow, all being the same commonsense reasoning tasks. In their experiments, PuDDing consistently achieved the highest mean accuracy among various pruning strategies when tested on zero-shot commonsense QA tasks at 9~20% layer sparsity​. For example, Table 4 shows PuDDing surpassing other baselines on ARC, HellaSwag, etc., after fine-tuning with LoRA. As for speed, they measure actual wall-clock time on GPUs: PuDDing yields about 1.21–1.25× speedup in different settings, with the router’s overhead being minimal. Overall, the evidence is convincing that PuDDing meets its goals: it yields a speedup and better accuracy than static pruning baselines, confirmed by multiple benchmarks and metrics.

**Essential References Not Discussed:**

It fairly covers most literature, but it might be good to cover a few more baselines such as LaCo, LLM-streamline, and FinerCut.

**Experimental Designs Or Analyses:**

The experiments and ablation studies are well-designed to isolate the benefits of PuDDing. The authors also test across different model architectures (LLaMA-based, Vicuna, OPT). The speed analysis is detailed as well – breaking down the time into pre-fill and generation phases along with router inference time. This gives a complete picture of how the method performs in practice.

**Methods And Evaluation Criteria:**

The paper focuses on accuracy on NLP benchmarks and actual inference speed. They evaluate on a suite of commonsense reasoning tasks (ARC-Easy/Challenge, HellaSwag, PIQA, WinoGrande, BoolQ), which are challenging tasks that can benefit from the full depth of a model. \
That said, the main drawback of the proposed method is construction of omission sets, while I acknowledge the authors already discussed this in the final section of the paper. The authors need to devise an experimental setting where omission sets largely differ, so that prompts drop notably different layers. There can be math, coding, medical, law tasks. From Figure 4, commonsense reasoning tasks seem to drop approximately the same layers which degrades the need for the proposed method. In case the omission sets overlap much, I wonder whether we can choose to have a single omission set per task via constructing a task-representation vector instead.

**Other Comments Or Suggestions:**

N/A

--------------------After Rebuttal--------------------

I have raised my score from 2 to 3, and lean towards acceptance, only if the authors faithfully include new experimental results in the final manuscript.

**Other Strengths And Weaknesses:**

N/A

**Questions For Authors:**

N/A

**Relation To Broader Scientific Literature:**

N/A.

**Theoretical Claims:**

N/A

---

> ### Author Rebuttal · Authors · 2025-03-31
>
> Dear reviewer 2qCi,
> Thank you for your insightful comments and suggestions. In what follows, we respond to the points raised by the reviewer one-by-one.
>
>
> ---
>
> ### **The scope of the task is too narrow, all being the commonsense reasoning tasks**.
> **TL;DR. We have already evaluated PuDDing on MMLU/OBQA/MathQA (Table 5), and newly added experiments on PubMedQA/SciQ.**
> First, we clarify that we have already evaluated PuDDing (trained on commonsense tasks) on specialized tasks. In particular, see Table 5 for results on MMLU/MathQA/OpenBookQA. From the table, we confirm that PuDDing generalizes well on these tasks as well.
> To make this point even more concrete, following the reviewer’s suggestion, we have added new experiments on PubMedQA/SciQ (see table below). Again, the results suggest that training PuDDing on commonsense tasks can give a competent router for specialized tasks. We hypothesize that this is because the optimal routing decision is not solely determined by the knowledge domain. Instead, there may be other notions of **diversity** inside commonsense reasoning tasks that affect how we should route, which also generalizes to specialized tasks.
>
> |LLaMA-3.1-8B|MMLU|MathQA|PubMedQA|sciQ|OpenbookQA|
> |:-|:-:|:-:|:-:|:-:|:-:|
> |Dense|63.49|39.53|75.80|96.00|44.60|
> |SLEB|23.76|25.19|56.40|89.20|36.00|
> |Shortened LLaMA|26.78|25.76|52.60|89.20|34.20|
> |**PuDDing**|**39.00**|**27.27**|**60.00**|**92.70**|**36.40**|
>
> ---
>
> ### **The main drawback of the proposed method is construction of omission sets (...) need to devise an experimental setting where omission sets largely differ, so that prompts drop notably different layers.**
>
> Our point is twofold:
> First, we clarify that despite the deceiving look of Figure 4, the actual omission sets differ quite notably for PuDDing trained with commonsense reasoning dataset. In the **[LINK](https://ibb.co/zTVMKrBL)**, we depict commonly omitted blocks for various tasks—including MMLU/MathQA/(…)—with explicit omission rates. We find that the omission rate of certain blocks differs dramatically over tasks. For instance, block 11 is dropped 99% in PIQA, but is dropped only 34% in MMLU; block 18 is dropped over 40% on PIQA and WinoGrande, but almost never in other tasks.
> Second, we have followed your suggestion to train a new version of PuDDing, where we construct omission sets using diverse domain data: math (MathQA), medicine (PubMedQA), science (SciQ), and commonsense reasoning (ARC-Easy, WinoGrande). The results are given in the table below; see newPuDDing. We observe that the average performance slightly increases over PuDDing, driven by the performance boosts in newly added datasets.
>
> | Method     | Average | Arc-C | Arc-E | BoolQ | HellaSwag | PIQA | WinoGrande | MathQA | PubMedQA | sciQ |
> |:-|:-:|:-:|:-:|:-:|:-:|:-:|:-:|:-:|:-:|:-:|
> | Dense| 73.42| 53.50 | 81.52 | 82.20 |78.81|79.98|73.40| 39.53|75.80|96.00|
> | PuDDing| 61.28| 41.47 | 67.09 | 62.02 |62.92| 73.94 |64.16|27.27|60.00|92.70|
> | newPuDDing | 62.37  | 41.38 | 67.26 | 67.37 |63.68| 73.07 |64.56| 29.58  |62.00| 92.40|
>
> ---
>
> ### **It fairly covers most literature, but it might be good to cover a few more baselines such as LaCo, LLM-streamline, and FinerCut.**
>
> Following the reviewer’s suggestion, we are working towards adding more baselines to our main table. In particular, we have already added LLM-Streamline results (see table below). As the method involves fine-tuning the base model, we compare it with the LoRA fine-tuned version of PuDDing; we find that PuDDing outperforms this baseline as well.
> Unfortunately, we could not compare against LaCo and FinerCut during the response period, as FinerCut does not provide any code, and LaCo has only released a very limited amount of codes (in ipynb). We are currently implementing the codes for these methods, and will be able to add it in the revised manuscript.
>
> | LLaMA-3.1-8B   | Average | Arc-C | Arc-E | BoolQ | HellaSwag | PIQA  | Winogrande |
> |:-|:-:|:-:|:-:|:-:|:-:|:-:|:-:|
> | Dense| 74.90| 53.50 | 81.52 | 82.20 | 78.81| 79.98 | 73.40|
> | LLM-Streamline (w/ fine-tune)| 66.08| 44.80 | 70.12 | 70.06 | 67.15| 72.63 | **71.74**|
> | PuDDing (w/ LoRA fine-tune)| **68.01**| **45.39** | **75.34** | **71.96** | **71.58** | **77.26** | 66.54|
> ---
> We sincerely hope that you find our response reasonable. Please don’t hesitate to let us know if there are any further questions.
>
> Best regards,
> Authors

---

> > ### Comment · Reviewer_2qCi · 2025-04-04
> >
> > I thank the authors for the additional results which resolved most of raised issues. I thus raise my score from 2 to 3, and lean towards acceptance, only if the authors faithfully include new experimental results in the final manuscript.

---

> > > ### Author Response · Authors · 2025-04-04
> > >
> > > Dear Reviewer 2qCi,
> > >
> > > Thank you for your positive response and for raising the score. We will incorporate all your suggestions into the manuscript.
> > > Please let us know if you have any further questions or concerns.
> > >
> > > Best regards,
> > > Authors

---

### Decision · Program_Chairs · 2025-05-01

**Decision:**

Accept (poster)

**Comment:**

The paper introduces PuDDing, that skips some of the layers during inference. The key idea is that the importance of a specific layer can be prompt dependent. Hence, PuDDing introduces a small and quick router model to decide which pre-selected layers to use based on the input. The simple and well-designed method is able to provide real speedups without needing special setups. The experiments also show that it outperforms removing the same layers all the time. However, the speedup (1.2-1.25x) was considered modest. Initial concerns about the narrowness of the tasks tested and lack of on-device benchmarks were addressed by the authors during rebuttal.